# Zenith: Real-time Identification of DASH Encrypted Video Traffic with Distortion

## ABSTRACT

Some video traffic carries harmful content, such as hate speech and child abuse, primarily encrypted and transmitted through Dynamic Adaptive Streaming over HTTP (DASH). Promptly identifying and intercepting traffic of harmful videos is crucial in network regulation. However, QUIC is becoming another DASH transport protocol in addition to TCP. On the other hand, complex network environments and diverse playback modes lead to significant distortions in traffic. The issues above have not been effectively addressed. This paper proposes a real-time identification method for DASH encrypted video traffic with distortion, named Zenith. We extract stable video segment sequences under various itags as video fingerprints to tackle resolution changes and propose a method of traffic fingerprint extraction under QUIC and VPN. Subsequently, simulating the sequence matching problem as a natural language problem, we propose Traffic Language Model (TLM), which can effectively address video data loss and retransmission. Finally, we propose a frequency dictionary to accelerate Zenith's speed further. Zenith significantly improves accuracy and speed compared to other SOTA methods in various complex scenarios, especially in QUIC, VPN, automatic resolution, and low bandwidth. Zenith requires traffic for just half a minute of video content to achieve precise identification, demonstrating its real-time effectiveness. The project page is available at https://anonymous.4open.science/r/Zenith-Anonymous.

## CCS CONCEPTS

• **Security and privacy** → **Web application security**; **Social network security and privacy**.

## KEYWORDS

Video Regulation, Encrypted Video Traffic, Traffic Identification, DASH, QUIC, Poor Network, Automatic Resolution

## 1 INTRODUCTION

Video traffic accounts for the most significant proportion of network traffic on the Internet currently. As of 2022, the proportion of video traffic to network traffic has reached 65.93%, an increase of 24% compared to 2021 [34]. In 2023, 85.9% of video traffic underwent encryption. Many harmful contents, such as hate speech and child abuse, are disseminated through encrypted video traffic [13], posing

*ACM MM, 2024, Melbourne, Australia*
© 2024 Copyright held by the owner/author(s). Publication rights licensed to ACM.
ACM ISBN 978-x-xxxx-xxxx-x/YY/MM
https://doi.org/10.1145/nnnnnnn.nnnnnnn

significant challenges to network regulation. Research on encrypted video traffic identification is of great significance to public safety.

Major video service providers like YouTube, Netflix, Hulu, and Amazon Prime have consistently used DASH [50] for video traffic transmission. To accelerate transmission speed, DASH has begun employing the QUIC protocol for transmitting video data. Most of the data packets in QUIC are fully authenticated and encrypted, and the multiplexing mechanism of QUIC [19] also poses many challenges. Most research on encrypted video traffic based on DASH is limited to TCP transport, with relatively little research on QUIC transport.

The complete video is divided into multiple video contents and encoded into multiple video chunks using Variable Bit Rate (VBR) [54] in DASH, then transmitted sequentially. VBR encoding dynamically allocates bit rates based on video content, resulting in a unique video chunk sequence for a specific video under a specific encoding. Previously, most research used video chunk sequences as video fingerprints, but it is now discovered that video chunk sequences are unstable [53]. The video chunk sequence for a specific video under a particular encoding scheme changes over time due to network conditions, significantly affecting the accuracy of encrypted video traffic identification.

Content Delivery Network (CDN) [21] has been consistently utilized to enhance the smoothness of video in DASH. When congestion occurs in the link between the client and the CDN node, it automatically switches to a CDN node with better network quality. Cross-transfer of video data across different streams may lead to loss [55]. Additionally, under severe network fluctuations, video data may be retransmitted [45]. The loss and retransmission of video data significantly affect the accuracy of encrypted video traffic identification, yet there is currently scarce research analyzing these two critical challenges.

Most studies only focus on the simplest playback modes. Playing videos under a VPN encapsulates video data with additional headers [39], causing significant noise in encrypted video traffic. Resolution switching also leads to severe distortion in encrypted video traffic [28]. Methods for identifying encrypted video traffic that can adapt to a broader range of video playback modes are more meaningful.

To tackle the above challenges, this paper proposes a real-time identification method for DASH encrypted video traffic with distortion called Zenith. We extract video segment sequences under various itags from video file headers as video fingerprints to construct the fingerprint database and extract the video chunk sequence from DASH encrypted video traffic transported by QUIC or TCP under VPN as the traffic fingerprint. Using the random walk model, we convert video fingerprints into video intermediate fingerprints in the form of video chunk sequences and then match the traffic fingerprint with each video intermediate fingerprint in the fingerprint database. We simulate the sequence matching problem as a natural language problem and propose TLM, which calculates the fingerprint similarity in two stages using the fingerprint sequence subset

frequency. The identification result is the video corresponding to the intermediate fingerprint with the highest similarity. Finally, we propose a frequency dictionary to accelerate Zenith's speed further. The overview of Zenith is shown in Figure 1. In various complex scenarios, we compare Zenith with other SOTA methods [2, 46, 49, 53]. Zenith improves accuracy ranging from 2.77% to 35.38%, with an average of 18.43%. Additionally, Zenith's speed can reach 9.87 μs. Zenith achieves 97.32% accuracy with traffic for just half a minute of video content, demonstrating its real-time effectiveness.

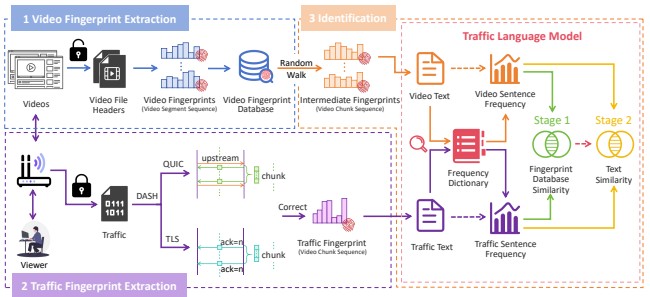

**Figure 1: The overview of Zenith.**

The main contributions of this paper are as follows:

- We design the fingerprint database, ensuring the stability and adaptability of video fingerprints across various resolutions.
- We extract the traffic fingerprint from DASH encrypted video traffic transported by QUIC or TCP under VPN, with errors within ±50.
- We propose TLM, significantly alleviating issues such as video data loss and retransmission. We additionally propose a frequency dictionary to accelerate Zenith's speed further.
- We compare Zenith with other SOTA methods in various complex scenarios. Zenith improves the accuracy by 18.43% on average and can reach 9.87 μs speed. Additionally, Zenith achieves 97.32% accuracy with traffic for just half a minute of video content.

The organization of this paper is as follows. Section 2 provides an overview of the current research progress in DASH encrypted video traffic identification. Section 3 elaborates on the process of Zenith, a real-time identification method for DASH encrypted video traffic with distortion. Section 4 conducts experimental evaluations on the accuracy, real-time performance, and robustness of Zenith from various aspects. Section 5 summarizes the research work of this paper.

## 2 RELATED WORK

Research on methods for identifying DASH encrypted video traffic begins with extracting meaningful features from encrypted video traffic [24]. Some studies focus on extracting time series features to construct fingerprints for matching. Others extract multidimensional features and then utilize deep learning models for training and identification.

### 2.1 Time Series Method

The video chunks mentioned in Section 1 are the mainstream time series features [12]. Matching video chunk sequences as fingerprints primarily employs time series methods. Using the differential concept, Gu et al. [10] proposed P-DTW for video fingerprint matching. Wu et al. [46] transformed fingerprints through differential transformation and aligned video fingerprints using a sliding window. Yang et al. [49] converted video fingerprints into tensor form using Markov chains and then matched them by comparing the similarity between tensors. Reed et al. [32] constructed a fingerprint database using kd-trees and calculated similarity based on the shape of the video fingerprint's time series for matching. However, it requires a significant distinction between video chunks within the window. Dubin et al. [6] used the peak burst bit rate of videos as video fingerprints. Stikkelorum et al. [41] employed finite state machines for matching, but the video chunk sequence must be continuous and severely affected by missing and retransmitted video chunks. Song et al. [40] extracted Media Presentation Description (MPD) files from encrypted video traffic through a man-in-the-middle attack and then extracted video fingerprints from MPD files. Zhang et al. [53] extracted video segment sequences from the file header as video fingerprints. Most of the above methods based on time series utilize Dynamic Time Warping (DTW) or its variants to match [42]. These methods perform effectively in small-scale applications as they do not rely on large datasets for training [30]. However, as the network environment and playback modes become complex and the scale of encrypted video traffic increases, the accuracy and speed of these methods are significantly affected.

### 2.2 Deep Learning Method

Methods based on deep learning extract multidimensional features from encrypted video traffic for training. Schuster et al. [35] extracted features from raw encrypted video traffic without processing and utilized Convolutional Neural Network (CNN) as the model for training. However, the model suffered significant distortion under frequent CDN switching. Khan et al. [16] used Bits Per Second (BPS) as a feature and employed Sequence Convolutional Neural Network (SCNN) to learn the changes in BPS. Lotfollahi et al. [25] still utilized video chunk sequences as features and trained using CNN, but the one-dimensional features limited the model's iterative updates. Li et al. [22] experimented with CNN, Recurrent Neural Network (RNN), and Multilayer Perceptron (MLP). Bae et al. [2] extracted Data Center Interconnect (DCI) information from encrypted video traffic input into CNN and then applied CNN to Long Term Evolution (LTE) networks for identification. With much training, deep learning models achieve high identification accuracy but require substantial data [33]. Furthermore, they are unsuitable for natural network environments because when new videos need to be added to the model, the entire model needs to be retrained, which requires significant resources.

## 3 METHODOLOGY

This section will elaborate on the core method Zenith of this paper. Zenith is mainly divided into two parts: fingerprint extraction and identification.

## 3.1 Fingerprint Extraction

In Zenith, two types of fingerprints need to be extracted. One is video fingerprints used to identify videos, which constitute the fingerprint database. The other is the traffic fingerprint extracted from encrypted video traffic, which serves as the identification target and is matched against video fingerprints in the fingerprint database. The following will detail the extraction methods for these two types of fingerprints.

*3.1.1 Video Fingerprint Extraction.* As mentioned in Section 1, video chunk sequences have been used as video fingerprints. However, it is now discovered that a video chunk is composed of several video segments. The variability in the combination of these video segments results in the instability of video chunk sequences [53]. Since video segment sequences are stable and invariant, we choose to use video segment sequences as video fingerprints.

*Video Segment Extraction.* MPD [29] file is a streaming media description file in DASH, which records various information about video transmission, including video segment information. However, video service providers no longer use MPD files. They embed this information into the headers of video files [53]. Video files come in two formats: fmp4 and webm. The Segment Index Box (sidx) and cues are control elements within fmp4 and webm, respectively, used to store relevant information about video segments and keyframes. This facilitates random access during streaming media playback and efficient data retrieval. The index values of video segment information in sidx and cues are located in the "adaptiveFormats" field of the web element, where the "start" and "end" values in the "indexRange" indicate the starting and ending indexes of video segment information. The "Reference_Size" in sidx and the difference between the "Cluster Position" in two adjacent cues represent the size of video segments. This enables the extraction of video segment sequences as video fingerprints directly from the video header file.

*Fingerprint Database Construction.* The video fingerprint uniquely corresponds to a specific video under a specific itag, where itag [4] represents a combination of specific media type (video/audio), file format (fmp4/webm), encoding method, resolution, frame rate, and other factors. Different resolutions of the same video correspond to different itags. To adapt to the playback mode of automatic resolution, when constructing the fingerprint database, video fingerprints corresponding to various itags with higher occurrence rates for each video are selected and added to the fingerprint database.

*3.1.2 Traffic Fingerprint Extraction.* Zenith uses stable video segment sequences as video fingerprints, but it is not feasible to extract video segment sequences from encrypted video traffic [36]. This is because the complete video is divided into multiple video contents in DASH, and each video content corresponds to a video chunk instead of a video segment. Therefore, only the video chunk sequence can be extracted as the traffic fingerprint.

*Video Chunk Extraction.* DASH encrypted video traffic by TCP transport is first divided into flows based on the five-tuple srcIP, srcPort, dstIP, dstPort, Protocol. Then, the encrypted video traffic is extracted from the traffic based on the Server Name Indication (SNI) in the TCP handshake [48]. For example, the SNI field in the encrypted traffic of YouTube and Netflix will contain "google-video.com" and "netflix.com". In encrypted video traffic, packets containing the same video or audio content have duplicate ACK numbers [5], and contents are transmitted sequentially. Therefore, the video chunk sequence can be extracted from the encrypted video traffic based on the ACK number.

However, QUIC does not have an SNI field and the three-way handshake process of TCP, so the video chunk sequence cannot be divided by the ACK number. For DASH encrypted video traffic by QUIC transport, traffic is divided into flows based on the five-tuple. According to statistics, flows larger than 4MB have a 91.9% probability of being encrypted video flows [8]. So, encrypted video traffic is extracted from the encrypted traffic by judging the flow size. Although one of the main features of QUIC is multiplexing, under CDN services, QUIC transmits audio and video content independently in sequence [43]. The client first sends multiple small and continuous request packets to the server, and then the server sends video content to the client. Video content consists of multiple 1250-byte data packets. Therefore, the sum of the downlink data packets sandwiched between two uplink data packets is the size of a video chunk, from which the video chunk sequence can be extracted from the encrypted video traffic.

*Offset Correction.* The video chunk extracted through the method mentioned above is larger than the actual video chunk because the extracted video chunk size includes the sizes of various headers encapsulated within the data packets. Encrypted data packets under TCP encapsulate HTTP headers and TLS headers [18], while encrypted data packets under QUIC encapsulate HTTP/3 headers and QUIC headers [38]. Additionally, if playing videos in a VPN, there will be an extra encapsulation of VPN headers [17]. The encapsulation of video data under both TCP and QUIC transmitting follows a similar pattern. Taking QUIC as an example, it is shown in Figure 2.

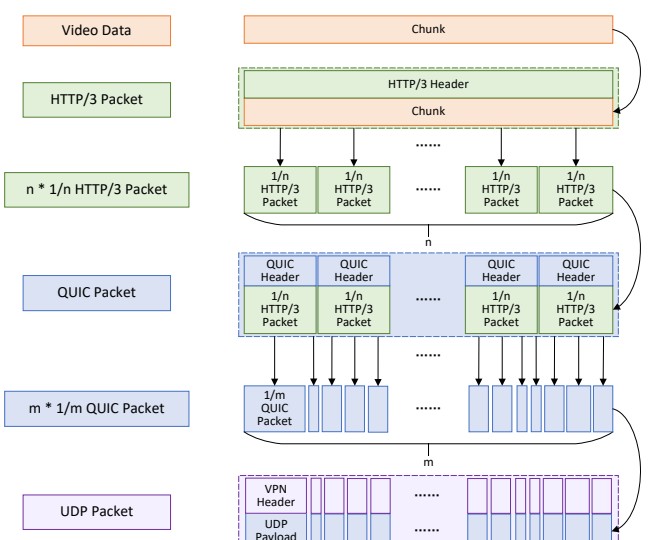

**Figure 2: Video data encapsulation and transmission process.**

The formulas can be obtained from Figure 2 as follows:

$$chunk + HTTP/3\_header = n \cdot QUIC\_payload \tag{1}$$

$$QUIC\_header + QUIC\_payload = QUIC\_packet \tag{2}$$

$$n \cdot QUIC\_packet = \sum_{i}^{m} UDP\_payload_i \tag{3}$$

$$m \cdot VPN\_header + \sum_{i}^{m} UDP\_payload_i = \sum_{j}^{m} UDP\_packet_j \tag{4}$$

$chunk$ refers to the actual size of the video chunk. An HTTP/3 packet is divided into $n$ QUIC packets, and $n$ QUIC packets are divided into $m$ UDP packets. Eliminating $n$, the formula for calculating the chunk size is as follows:

$$chunk = \left( \sum_{j}^{m} UDP\_packet_j - m \cdot VPN\_header \right)$$
$$\cdot \frac{QUIC\_payload}{QUIC\_packet} - HTTP/3\_header \tag{5}$$

$QUIC\_payload$, $QUIC\_packet$, and $HTTP/3\_header$ are fixed values in DASH [27], while $VPN\_header$ remains constant within the same VPN service, with variations in VPN encapsulation headers between different VPN services not exceeding 10 bytes [44]. $\sum_{j}^{m} UDP\_packet_j$ is equivalent to the size of the video chunk directly extracted from encrypted video traffic, denoted as $tra\_extra$. In summary, there exists a linear relationship between $chunk$ and $tra\_extra$, and the simplified formula is as follows:

$$chunk = \alpha \cdot tra\_extra - \beta \tag{6}$$

Section 4.2.1 discusses the optimal values for parameters $\alpha$ and $\beta$. After the correction operations, video chunks that closely approximate actual video chunk sizes can be obtained. Traffic fingerprints contain both video and audio chunks. Audio chunks, significantly smaller than video chunks and with less discriminative power, are unsuitable as fingerprint elements [9]. As shown in Figure 3, the size ranges of video chunks and audio chunks hardly overlap, allowing for filtering audio chunks based on a threshold. Chunks below 600KB are considered audio chunks [51]. Furthermore, only video fingerprints corresponding to itags labeled "video" are selected when constructing the fingerprint database.

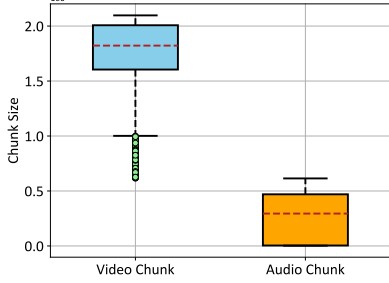

**Figure 3: Video and audio chunk size box.**

## 3.2 Identification

### 3.2.1 Video Fingerprint Transformation.
The fingerprint database constructed by video fingerprints serves as the identification template, while the traffic fingerprint serves as the identification target. However, since video fingerprints are in the form of video segment sequences and the traffic fingerprint is in the form of video chunk sequences, direct matching operations cannot be performed. As mentioned in Section 3.1.1, a video chunk is composed of multiple video segments, so video fingerprints can be transformed by combining video segments to form video chunk sequences, referred to as video intermediate fingerprints. Two rules for combining video segments were obtained through experimentation. **Rule 1**: Video segments tend to be combined as much as possible until the maximum size can be combined [3]. This is because the maximum size of a video chunk is 2MB, and statistics show that 95.7% of video chunks would exceed this maximum value if combined with the following video segment. **Rule 2**: Combining the first few video segments is relatively random [14]. An error in one combination point will significantly affect subsequent combinations. Although video segments of a video fingerprint can simulate many combinations, we plan to simulate only one most suitable combination method for algorithm speed.

Random walk is commonly used to characterize irregular variations, and a one-dimensional random walk model can also be viewed as a Markov chain. When the walk reaches a steady state, the probability of visiting each node conforms to a specific distribution [15]. Taking the video fingerprint as a sequence of walks, following **Rule 1** and **Rule 2** during walks, there is also **Rule 3**: The walk is unidirectional backward. Assuming a video fingerprint $VF = \{s_1, s_2, ..., s_n\}$, where $s$ represents a video segment, and a single walk result is $RW = \{r_1, r_2, ..., r_n\}$, where $r_i = 1$ indicates that the $i$-th video segment is visited during this walk, and $r_i = 0$ indicates that it is not visited. Assuming a total of $m$ iterations of walks, the frequency of the $i$-th video segment being visited is calculated as follows:

$$fre_i = \frac{\sum_{j}^{m} r_i \left( r_i \in RW_j \right)}{\sum_{l}^{n} \sum_{k}^{m} r_l \left( r_l \in RW_k \right)} \tag{7}$$

where $RW_j$ represents the result of the $j$-th walk. After $m$ iterations of walks, the walk frequency vector of the video fingerprint is denoted as $Fre^m = (fre_1, fre_2, ..., fre_n)$. After iterating walks, it begins to iterate probability. The walk frequency vector is the initial walk probability vector $P_0 = Fre^m$. The formula for the walk probability vector for the $t$-th iteration is as follows:

$$P_t = (1 - \theta) \cdot P_0 + \theta \cdot P_{t-1} \tag{8}$$

where $\theta$ is the damping factor, representing the probability of selecting a forward walking strategy. Zenith does not allow backward walking, so $1 - \theta$ represents the probability of staying in the same position. After multiple iterations of walk probability, the probability of each video segment being walked will stabilize [47]. The final walk probability vector $P_m = (p_1, p_2, ..., p_n)$ is subjected to a sigmoid transformation in the following formula:

$$u_i = \begin{cases} 0, & \frac{1}{1 + e^{-p_i}} \le 0.5 \\ 1, & \frac{1}{1 + e^{-p_i}} > 0.5 \end{cases} \tag{9}$$

$u_i$ is the decision factor, where $u_i = 1$ indicates that the $i$-th video segment is the endpoint of a combination. The final decision vector is denoted as $U = (u_1, u_2, ..., u_n)$. Combine the video segments enclosed between adjacent decision factors with a value of 1. For example, if $u_4 = 1, u_5 = 0, u_6 = 0, u_7 = 1$, then the 5th, 6th, and 7th video segments should be combined into one video chunk. Using this method, all video fingerprints in the fingerprint database are converted into video intermediate fingerprints.

*3.2.2 Traffic Language Model.* The matching between fingerprints is the matching between video chunk sequences. As mentioned in Section 1, adverse network conditions can result in the loss and retransmission of video chunks, severely disrupting the continuity of the traffic fingerprint, leading to a significant decrease in matching accuracy. Therefore, designing an algorithm that does not rely on sequence continuity is necessary. This paper proposes TLM, which transforms the sequence matching problem into a natural language problem. TLM utilizes the fingerprint sequence subset frequency to bypass the continuity being disrupted. First, regarding video chunks as words, fingerprints as texts, and fingerprint sequence subsets as sentences.

*Chunk to Word.* Assuming the fingerprint database $FD = \{v_1, v_2, ...\}$ contains numerous video fingerprints $VF_v^{itag} = \{s_1, s_2, ...\}$, where $VF_v^{itag}$ represents the video fingerprint of video $v$ under a certain $itag$, and $s$ denotes a video segment. After the transformation of video fingerprints described in Section 3.2.1, video intermediate fingerprints $MF_v^{itag} = \{c_1, c_2, ...\}$ are obtained, where $c$ represents a video chunk. Capturing traffic at the gateway, the traffic fingerprint $TF = \{c_1, c_2, ...\}$ is extracted following the approach outlined in Section 3.1.2. To convert video chunks into words, the size range of video chunks is initially divided into multiple areas of equal size, with the partitioning formula as follows:

$$area = \frac{size_{max} - size_{min}}{area\_num} \quad (10)$$

where $area$ represents the size of the area, $size_{max}$ is the maximum value of the video chunk, typically 2MB [52], $size_{min}$ is the minimum value of the video chunk, typically 600KB [51], and $area\_num$ denotes the number of areas. The optimal value for $area\_num$ is discussed in Section 4.2.2. Each video chunk is assigned to its corresponding area according to the following formula:

$$sn = \lceil \frac{c - size_{min}}{area} \rceil \quad (11)$$

The video chunk $c$ is assigned to the $sn$-th area, thus transforming the video chunk into a word, denoted as $W^{sn}$. The traffic fingerprint $TF$ is transformed into a traffic text $TT = \{w_1, w_2, ...\}$, and video intermediate fingerprints $MF_v^{itag}$ are transformed into video texts $MT_v^{itag}$.

*Sentence Frequency.* Utilizing the entire fingerprint for matching operations would impose a significant computational burden on the algorithm [20]. Hence, matching is commonly performed using subsets of fingerprint sequences, with shorter subset lengths yielding faster algorithms. Fingerprint sequence subsets correspond to sentences in TLM, the sentence set denoted as $Sen = \{sen_1, sen_2, ...\}$, where each sentence has the same length. For instance, if the sentence length is set to 2, the sentence set becomes $Sen = \{w_1 w_2, w_2 w_3, ...\}$.

Section 4.2.2 discusses the optimal value for the sentence length $sen\_len$. The same sentence can appear multiple times in a text, and $ct_{sen}^T$ is used to record the number of sentences $sen$ in text $T$.

In a typical language model, the text similarity is often analyzed by computing word frequency [31]. Zenith analyzes the fingerprint similarity from both macro and micro perspectives. Firstly, from a macro perspective, the frequency of each sentence in the fingerprint database is calculated as shown in the following formula:

$$FD\_Fre_{sen} = \log \left( \frac{\sum_{v \in FD} \sum_{sen_i \in MSen_v^{itag}} ct_{sen_i}^{MT_v^{itag}}}{\sum_{v \in FD} ct_{sen}^{MT_v^{itag}} + 1} \right) \quad (12)$$

$MSen_v^{itag}$ is the sentence set of $MT_v^{itag}$, $\sum_{v \in FD} \sum_{sen_i \in MSen_v^{itag}} ct_{sen_i}^{MT_v^{itag}}$ represents the number of total sentences in the fingerprint database, and $\sum_{v \in FD} ct_{sen}^{MT_v^{itag}}$ represents the number of $sen$ in the fingerprint database. The reason for taking the reciprocal logarithm of the frequency is that when the frequency of a sentence in the fingerprint database is lower, it indicates that the sentence is unique and more representative and should be given higher importance during the matching process. The frequency in the fingerprint database reflects the macro weight of a sentence. Then, from a micro perspective, the frequency of each sentence in a particular text is calculated as shown in the formula:

$$T\_Fre_{sen}^T = \frac{ct_{sen}^T}{\sum_{sen_i \in Sen} ct_{sen_i}^T} \quad (13)$$

$Sen$ is the sentence set of $T$, and $\sum_{sen_i \in Sen} ct_{sen_i}^T$ reprsents the number of total sentences in $T$.

*Fingerprint Similarity.* Next, sentence frequency calculates the fingerprint similarity in two stages based on macro and micro perspectives. The first stage utilizes sentence frequency in the fingerprint database from a macro perspective to compute the fingerprint similarity between the video intermediate fingerprint of video $v$ and the traffic fingerprint at the fingerprint database dimension, as shown in the following formula:

$$FD\_Sim_v = \sum_{sen \in TSen \cap MSen_v^{itag}} FD\_Fre_{sen} \quad (14)$$

where $TSen$ is the sentence set of $TT$. As mentioned above, the sentence frequency in the fingerprint database represents the weight of the sentence in the fingerprint database. Here, the weight of the shared sentences between the two fingerprint texts is directly used as the fingerprint similarity criterion. This matching approach has lower complexity, significantly improving the speed.

However, the above strategy also reduces the discriminative power of the matching, potentially resulting in multiple video intermediate fingerprints having the same maximum similarity with the traffic fingerprint. In such cases, the second stage of matching is initiated. This stage utilizes sentence frequency in the text from a micro perspective to compute the fingerprint similarity between the video intermediate fingerprint of video $v$ and the traffic fingerprint at the text dimension, as shown in the following formula:

$$T\_Sim_v = \sum_{sen \in TSen \cap MSen_v^{itag}} \left( T\_Fre_{sen}^{TT} \cdot T\_Fre_{sen}^{MT_v^{itag}} \right) \quad (15)$$

The text similarity accumulates the product of frequency in the text of sentences shared between the two fingerprint texts. Finally, the identification result is the video corresponding to the video intermediate fingerprint with the highest similarity to the traffic fingerprint. An example is simulated to facilitate understanding, as shown in Figure 4.

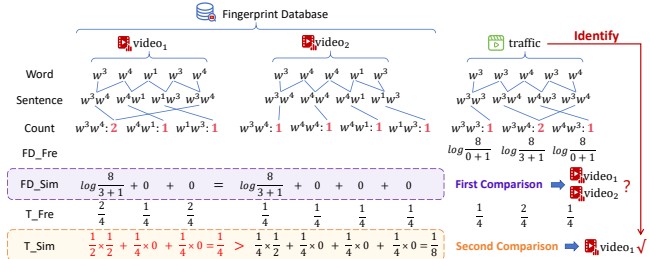

**Figure 4: Example of the TLM process.**

*Frequency Dictionary.* We propose a dictionary to store sentence frequency information to accelerate the process further. The structure is illustrated in Figure 5. The frequency dictionary records the occurrence frequency of each sentence in both the fingerprint database and texts. Let $sen\_num$ denote the number of sentences in the traffic fingerprint, and $fp\_num$ denote the number of video fingerprints in the fingerprint database. The time complexity is $O(sen\_num + sen\_num \cdot fp\_num + fp\_num)$. However, since $fp\_num$ is significantly larger than $sen\_num$, the time complexity is ultimately simplified to $O(fp\_num)$. Whenever a new video is added to the database, only minor changes to the fingerprint database and frequency dictionary are required. This offers significant advantages over deep learning methods, which require retraining [37].

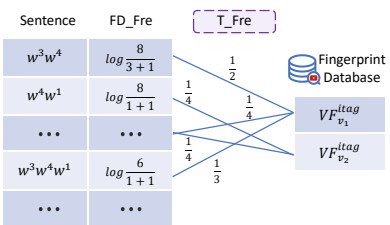

**Figure 5: Frequency dictionary structure.**

## 4 EVALUATION

### 4.1 Dataset

Platforms like Netflix, which offers movies and TV shows, undergo strict scrutiny by video service providers before releasing content. However, platforms like YouTube, where individuals upload videos, are more prone to hosting harmful content [1]. Therefore, this study primarily collects data from the YouTube platform. The experiment involved 5186 videos since 2021, covering ten categories such as sports, food, and travel. Video fingerprints are collected for each video under 12 different itags to build the fingerprint database,

**Table 1: Itag information.**

| Itag | Resolution | FPS | Itag | Resolution | FPS |
|------|-----------|-----|------|-----------|-----|
| 134 | 360p | 30 | 136 | 720p | 30 |
| 396 | 360p | 30 | 398 | 720p | 60 |
| 243 | 360p | 30 | 247 | 720p | 30 |
| 135 | 480p | 30 | 137 | 1080p | 30 |
| 397 | 480p | 30 | 399 | 1080p | 60 |
| 244 | 480p | 30 | 248 | 1080p | 30 |

**Table 2: Experimental scenario information.**

| Scenario | Connection | Resolution | Bandwidth | VPN |
|----------|-----------|-----------|-----------|-----|
| Ideal | wired | 720p | 5Mbps | none |
| Low Resolution | wired | 360p | 5Mbps | none |
| Auto Resolution | wi-fi | auto | 2Mbps | none |
| VPN | wired | 720p | 5Mbps | vpn |
| Low Bandwidth | wired | 720p | 500Kbps | none |

**Table 3: Amount of data in five scenarios.**

| Scenario | Stream Number | | Video Chunk Number | |
|----------|-------|-------|---------|---------|
| | TCP | QUIC | TCP | QUIC |
| Ideal | 5427 | 5531 | 515565 | 530976 |
| Low Resolution | 5264 | 5396 | 184242 | 194256 |
| Auto Resolution | 6232 | 6110 | 529727 | 525461 |
| VPN | 5843 | 5698 | 555085 | 547008 |
| Low Bandwidth | 7198 | 7233 | 539853 | 549708 |
| Sum | 29964 | 29968 | 2324472 | 2347409 |
| | 59932 | | 4671881 | |

resulting in 62232 video fingerprints. The corresponding resolutions and frame rates for each itag are listed in Table 1.

In order to simulate natural network environments and playback modes as closely as possible and to comprehensively measure Zenith's performance in various situations, this study designed five scenarios, as shown in Table 2. In these five scenarios, traffic transmitted within 10 minutes was collected for the 5186 videos mentioned above under both TCP and QUIC transport in DASH, and traffic fingerprints were extracted. A video may be transmitted through multiple streams [7]. The amount of data in five scenarios is presented in Table 3.

### 4.2 Parameter Adjustment

*4.2.1 $\alpha$ and $\beta$.* In Section 3.1.2, it is demonstrated that the sizes of video chunks extracted directly from encrypted video traffic are linearly related to the actual size of video chunks, as shown in Equation 6. The sizes of 4671881 pairs of extracted video chunks and actual video chunks from five scenarios are projected onto the coordinate axis and then fitted with a linear curve, as shown in Figure 6.

The slope of the resulting curve is 0.986317, and the intercept is -1123.45. Thus, $\alpha$ is 0.986317, and $\beta$ is 1123.45. Equation 6 can be rewritten as follows:

$$chunk = 0.986317 \cdot tra\_extra - 1123.45 \tag{16}$$

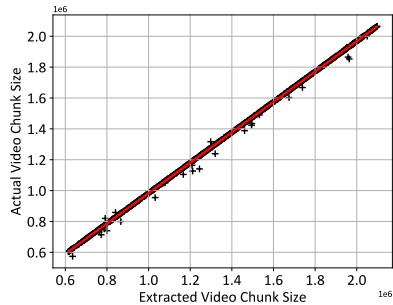

**Figure 6: Video chunks projection and fitting curve.**

The range of the difference between the corrected video chunk size and the actual video chunk size is only ±50, which has a negligible impact.

*4.2.2 area_num and sen_len.* In TLM, the number of areas denoted as *area_num* determines the granularity of the segmentation. With a higher *area_num*, different video chunks are likelier to be assigned to different words, resulting in greater distinguishability between video chunks. However, this also increases the computational burden of the algorithm. The sentence length denoted as *sen_len* influences the uniqueness of the sentences: longer sentences result in fewer identical sentences and greater uniqueness. However, longer sentences are more vulnerable to changes in resolution, loss and retransmission of video chunks. Set *sen_len* from 5 to 9 and explore the impact of *area_num* on Zenith's accuracy using datasets from automatic resolution and low-bandwidth scenarios. The results are shown in Figure 7.

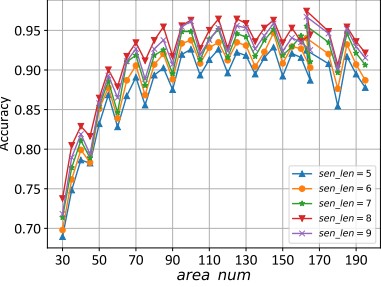

**Figure 7: Accuracy with different *area_num* and *sen_len*.**

The experimental results indicate that Zenith achieves the highest accuracy of 97.32% when *area_num* is set to 163 and *sen_len* is set to 8. A sentence length of 8 implies the need for only eight video chunks, each containing approximately 4 seconds of video content [11]. Consequently, Zenith can achieve excellent performance with traffic for just half a minute of video content.

## 4.3 Ablation

This paper designs TLM to operate in two stages from both macro and micro perspectives to accelerate speed and accuracy further. To verify the necessity of the macro stage, the matching is only in the second stage, conducted directly by computing the sentence frequency and fingerprint similarity in texts. Compared with TLM's complete two-stage identification, the experimental results are shown in Figure 8.

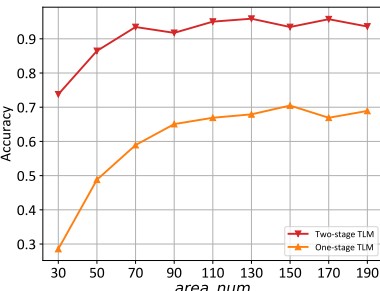

**Figure 8: Comparison of accuracy between two-stage TLM and one-stage TLM.**

The experimental results indicate that directly computing the sentence frequency and fingerprint similarity in texts yields an accuracy approximately 25.16% lower than the complete TLM, with an additional time requirement of approximately 10 ms. Calculating the fingerprint similarity at the fingerprint dimension is relatively straightforward. If results can be identified in the first stage, the fingerprint similarity at the text dimension computation becomes unnecessary, leading to significant time savings. The identification in the two stages involves mutual adjustments, which can further enhance accuracy.

## 4.4 Comparison

We compared Zenith with four other SOTA encryption video traffic identification methods. Because deep learning methods require retraining the entire model when new videos need to be added, they are unsuitable for real-time online applications. Therefore, only one higher accuracy deep learning method was selected for comparison, while the other three methods are time series related methods. The details of these baseline methods are introduced in Section 2. Experiments were conducted in the five scenarios mentioned in Section 4.1, and the experimental results are illustrated in Figure 9. Specific data information is presented in the Appendix.

Analysis of the experimental results leads to the following conclusions:

- Under low resolution, the traffic fingerprint is shorter [26]. When the video duration is below 120s, the accuracy of the other three time series methods decreases by 2.77% to 7.80% compared to the ideal scenario. In Zenith, TLM utilizes the fingerprint sequence subset frequency to calculate the fingerprint similarity, mitigating the impact of low resolution. The decrease in accuracy of Zenith is within 2.10%, and its accuracy is, on average, 13.63% higher than other methods.
- Network stability is lower in Wi-Fi environments, often leading to resolution switches [23]. Following a resolution switch, the size of video chunks within the traffic fingerprint undergoes significant distortion. The video fingerprints utilized by Wu and Yang are both unstable, while the CNN trained by Bae lacks generalization. Consequently, the accuracy of these three methods decreases by 4.78% to 12.76% compared

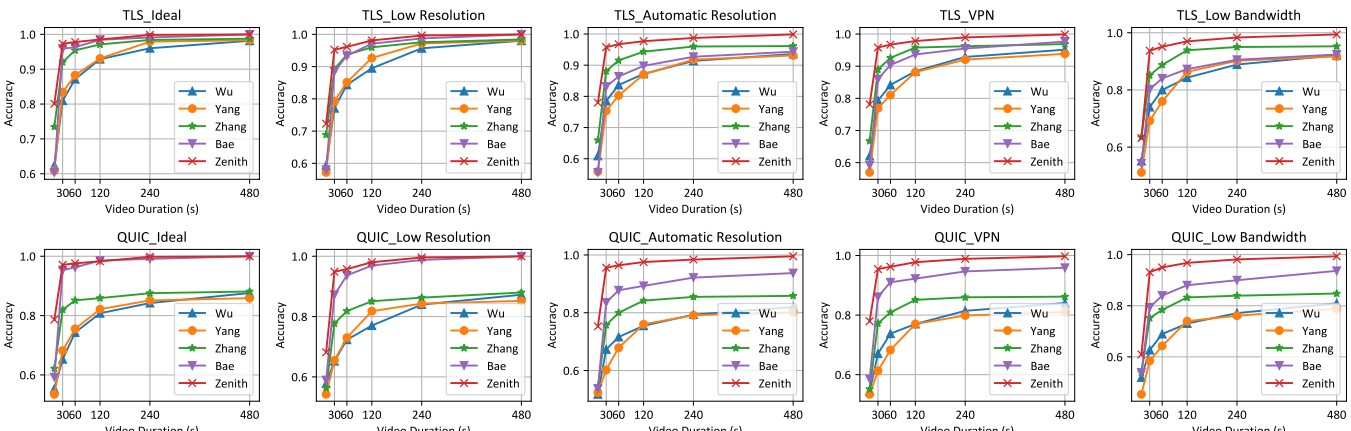

Figure 9: Accuracy of methods in five scenarios.

to the ideal scenario. Zenith constructs the fingerprint database using stable video segment sequences under various itags as video fingerprints, alleviating the impact of resolution switches. The decrease in accuracy of Zenith is within 2.12%, and its accuracy is, on average, 19.34% higher than other methods.

- When utilizing a VPN, original video data undergoes encapsulation with additional VPN headers. Wu, Yang, and Zhang overlook the alterations VPN imposes on traffic, resulting in a decrease in accuracy ranging from 3.02% to 9.89% compared to the ideal scenario. Zenith simulates the encapsulation of VPN headers when extracting the traffic fingerprint. The decrease in accuracy of Zenith is within 2.12%, and its accuracy is, on average, 17.61% higher than other methods.
- Low-bandwidth network environments often cause video data loss and retransmission, disrupting the traffic fingerprint's continuity. The other four methods heavily rely on the continuity of the traffic fingerprint, resulting in a decrease in accuracy ranging from 8.55% to 16.92% compared to the ideal scenario. Within Zenith, TLM employs the fingerprint sequence subset frequency as its computational core, independent of continuity. The decrease in accuracy of Zenith is within 3.60%, and its accuracy is, on average, 23.14% higher than other methods.
- The distinction between DASH video traffic transported with QUIC and TCP is significant. Methods for extracting the traffic fingerprint tailored to TCP cannot be similarly applied to QUIC, resulting in a decrease in accuracy ranging from 7.07% to 15.22% for the other methods on QUIC. Zenith effectively identifies QUIC encrypted video traffic in DASH with an accuracy approximately 11.56% higher than the SOTA methods.

The advantages and disadvantages of methods in five scenarios are shown in Table 4. In particular, although the deep learning method performs well in accuracy, it cannot be applied to natural networks due to the difficulty of updating the video database. Across the five scenarios, Zenith improves accuracy compared to the other four methods, ranging from 2.77% to 35.38%, with an

Table 4: The influence of methods in five scenarios.

| Method | Low Resolution | Auto Resolution | VPN | Low Bandwidth | QUIC |
|---|---|---|---|---|---|
| Wu | - | - | - | - | - |
| Yang | - | - | - | - | - |
| Zhang | - | + | - | - | - |
| Bae | + | - | + | - | + |
| Zenith | + | + | + | + | + |

+ represents that the method has advantages in the scenario.
- represents that the method has disadvantages in the scenario.

average of 18.43%. Additionally, Zenith's speed can reach 9.87 µs. In terms of real-time performance, Zenith achieves 97.32% accuracy with traffic for just half a minute of video content. In summary, Zenith is highly effective for QUIC and VPN, and maintains high accuracy and speed in automatic resolution playback mode and low bandwidth network environments. Zenith can be deployed in natural network environments for real-time DASH encrypted video traffic identification and regulation.

## 5 CONCLUSION

To address the challenges posed by QUIC transport, complex network environments, and diverse playback modes causing significant distortion in encrypted video traffic in DASH, this paper proposes Zenith. We extract stable video segment sequences under various itags as video fingerprints to tackle resolution changes and propose a method of traffic fingerprint extraction aiming at QUIC and VPN. The proposed TLM does not rely on sequence continuity and effectively addresses video data loss and retransmission. Finally, we propose a frequency dictionary to accelerate Zenith's speed further. Across various complex scenarios, Zenith significantly improves accuracy and speed compared to other SOTA methods. Additionally, Zenith achieves precise identification with traffic for just half a minute of video content, demonstrating its real-time effectiveness. Future research will focus on defense strategies against various encryption video traffic identification methods.

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
