# OpenReview forum: "Zenith: Real-time Identification of DASH Encrypted Video Traffic with Distortion"
_acmmm.org/ACMMM/2024/Conference — MM2024 Poster_

### Official Review · Reviewer_4BkX · 2024-05-23

**Rating:** 4
**Confidence:** 3

**Summary:**

This paper focused on the problem of identifying DASH-encrypted video traffic. Its goal is to develop an effective method for real-time identification of such traffic, addressing issues posed by QUIC transport, complex network environments, and diverse playback modes. The authors proposed a method called Zenith, which includes the Traffic Language Model (TLM) and a frequency dictionary to accelerate the process. To achieve this, the authors implement the system to extract stable video segment sequences as video fingerprints and match them with traffic fingerprints using a natural language processing approach. The authors compare Zenith with other methods in various complex scenarios, and results show that Zenith improves accuracy and achieves high speed, making it effective for real-time identification.

**Strengths:**

* The topic of identifying DASH-encrypted video traffic is interesting.
* The design sounds reasonable.
* The formulation is clear and easy to understand.

**Limitations:**

* The background information about QUIC is insufficient. Given the importance of QUIC in the context of identifying DASH-encrypted video traffic, a more thorough explanation of its characteristics and advantages is necessary. This would provide readers with a clearer understanding of the context and significance of the proposed method within the scope of current network protocols.
* The methods of fingerprint extraction are not novel.
* The experiment setup includes 1080p. However, the ideal scenario has the max resolution of 720p. It is not reasonable. This discrepancy raises concerns about the realism and applicability of the evaluation, as it does not accurately reflect common usage scenarios.
* The definition of evaluation metric accuracy is not clear.
* Need more fundamental and details explanations of the improvement compared to other baseline.

**Suitability:**

3

---

### Official Review · Reviewer_BTCe · 2024-05-23

**Rating:** 5
**Confidence:** 3

**Summary:**

This paper proposes a real-time identification method for DASH encrypted video traffic with distortion. The author proposes the fingerprint extraction methods and design the fingerprint database, ensuring the stability and adaptability of video fingerprints across various resolutions. They propose TLM, significantly alleviating issues such as video data loss and retransmission. They additionally propose a frequency dictionary to accelerate Zenith’s speed further.

**Strengths:**

(1)	Zenith's design is very comprehensive and reasonable. And the division of its modules and functions is very clear.
(2)	The theoretical basis for the proposed Zenith is sufficient. The author provides rigorous mathematical formula derivation.
(3)	The experimental evaluation data is very sufficient. First, the basis of parameter adjustment is given, and then the novelty of the scheme is verified.

**Limitations:**

(1)	The author mentions that deep learning methods have poor generalization ability. Is there more experimental data to support this conclusion?
(2)	The author claims that the dataset contains ten different types of videos. For different types of videos, Is Zenith's performance stable? How does its variance range compare to other schemes?
(3)	What is the identification accuracy of Zenith and other schemes when encrypted video traffic has a defense strategy against identification?

**Suitability:**

3

---

### Official Review · Reviewer_dVL7 · 2024-06-04

**Rating:** 5
**Confidence:** 4

**Summary:**

This paper addresses the encrypted video traffic identification issue. The author construct a video fingerprint database, and extract traffic fingerprint from QUIC and VPN traffic. A traffic language model is proposed to match the fingerprints between the database and the traffic, avoid the sequence disorder due to retransmission or packet loss. It leverages the frequency of fingerprint sequence subset as the similarity features to bypass the disrupted sequence.

**Strengths:**

1, the paper proposes a frequency-based sequence matching method, like text similarity matching by word frequency in natural language model. This method avoids the noise of disrupted video sequence due to retransmission or losses.

2, A fingerprint database is constructed with various resolution, frame-rate. A QUIC or VPN video traffic fingerprint extraction method is also proposed.

**Limitations:**

1, the paper presentation should be improved significantly. Some expressions are vague, such as Line-144 "stability and adaptability", line-247"video segment sequences are stable and invariant", Line-408"Video fingerprint transformation"

2, In Sec.3.1, the features to construct fingerprint database should be listed in a table. The traffic features extracted from QUIC or VPN should be also listed clearly.

3, The random walk to combine video segment sequence into chunk sequence determines the frequency of video segment. Why it can match  with the database? Please clarify the reason you use random walk to generate features of segment.

**Suitability:**

3

---

### Official Review · Reviewer_gUEK · 2024-06-09

**Rating:** 2
**Confidence:** 3

**Summary:**

This paper addresses the challenge of identifying harmful video content, such as hate speech and child abuse, transmitted through encrypted Dynamic Adaptive Streaming over HTTP (DASH) and QUIC protocols. Current methods struggle with complex network environments and diverse playback modes, causing significant traffic distortions. To solve this, the authors propose Zenith, a real-time identification method that extracts stable video segment sequences as fingerprints to handle resolution changes. Zenith uses a Traffic Language Model (TLM) to mitigate issues like data loss and retransmission and employs a frequency dictionary to enhance speed. Zenith outperforms other state-of-the-art methods in accuracy and speed, particularly in complex scenarios involving QUIC, VPN, and low bandwidth, needing only 30 seconds of video traffic for precise identification.

**Strengths:**

- The motivation and contribution are clear.
- This paper is well written and organised.
- This paper has a good presentation of the results.

**Limitations:**

- The topic does not have strong relevance to MM conference. It can be seen in the References section.
- The scenario considered in this paper where the MPD files are replaced by the headers of the video files would be limited to Youtube. Thus, a clear statement about where the proposed approach can be considered and used should be mention.
- In the evaluation section, other common resolutions  are not considered like 1440p, 4K are not considered.
- In Fig. 7, an explanation about lower accuracies of 60, 75, 90, 105,... is needed.
- The unit of axises in Fig. 6 is missing.
- The titles of references should be synchronised in a same format (capitalised or non-capitalised)

**Suitability:**

2

---

### Meta-Review · Area_Chair_AGsF · 2024-07-02

**Recommendation:** Accept (Poster)
**Confidence:** 4

**Metareview:**

As per the reviewer's feedback:

Quality:

The paper demonstrates a good quality of research and presentation. It proposes a frequency-based sequence matching method, similar to text similarity matching by word frequency in natural language models. This method is innovative in its approach to avoiding the noise of disrupted video sequences due to retransmission or losses. Additionally, the construction of a fingerprint database with various resolutions and frame rates, along with a QUIC or VPN video traffic fingerprint extraction method, is a good contribution to the field. However, the presentation of the paper could benefit from improvement. Some expressions are vague and could be clarified for better understanding. Moreover, the features to construct the fingerprint database and the traffic features extracted from QUIC or VPN should be listed clearly.

Clarity:

The paper is well written and organized, with a good presentation of results. The design and theoretical basis of the proposed method are clear and comprehensive. The division of modules and functions in the proposed Zenith system is clear, and the mathematical derivation is rigorous. However, there are some areas where the clarity could be improved:
* The explanation of lower accuracies in Fig. 7 needs to be provided.
* The units of the axes in Fig. 6 are missing.
* The titles of references should be synchronized in the same format (capitalized or non-capitalized).
* The background information about QUIC is insufficient. A more thorough explanation of its characteristics and advantages would provide readers with a clearer understanding of the context and significance of the proposed method.

Originality:

The originality of the paper lies in its method for sequence matching and the construction of a comprehensive fingerprint database. The approach to avoid noise in video sequences and the specific focus on QUIC and VPN traffic are novel contributions. However, the methods of fingerprint extraction are not entirely novel, which slightly reduces the paper's overall originality.

Significance:

The paper makes a good contribution to the field of multimedia systems and streaming. The proposed method and the construction of the fingerprint database have potential applications in identifying DASH-encrypted video traffic, which is a relevant and timely topic. However, the paper's relevance to the MM conference is moderate, as indicated by the references section and the specific scenario considered.

Pros:

* The motivation and contribution are clear.
* The paper is well written and organized and It presents results effectively.
* The frequency-based sequence matching method is innovative and effective.
* The construction of a fingerprint database with various resolutions and frame rates is a good contribution.
* The design and theoretical basis of Zenith are comprehensive and reasonable.
* Experimental evaluation data is sufficient and well-presented.

Cons:

* The scenario considered is limited to YouTube.
* Other common resolutions (e.g., 1440p, 4K) are not considered in the evaluation section.
* Lower accuracies in Fig. 7 need explanation.
* Units of axes in Fig. 6 are missing.
* Titles of references need synchronization.
* Some expressions are vague and need clarification.
* Background information about QUIC is insufficient.
* The methods of fingerprint extraction are not novel.
* The experiment setup's resolution discrepancy raises concerns about the realism of the evaluation.
* The definition of the evaluation metric accuracy is unclear.

Final Rating Justification:

The authors effectively addressed the concerns raised in the initial review, providing clarity and resolving ambiguities. The paper's contributions are clear and enough to warrant acceptance. Therefore, the final rating is Accept (Poster).